# Periodic Boosters of COVID-19 Vaccines Do Not Affect the Safety and Efficacy of Immune Checkpoint Inhibitors for Advanced Non-Small Cell Lung Cancer: A Longitudinal Analysis of the Vax-On-Third Study

**DOI:** 10.3390/cancers17121948

**Published:** 2025-06-11

**Authors:** Agnese Fabbri, Enzo Maria Ruggeri, Antonella Virtuoso, Diana Giannarelli, Armando Raso, Fabrizio Chegai, Daniele Remotti, Carlo Signorelli, Fabrizio Nelli

**Affiliations:** 1Medical Oncology Unit, Department of Oncology and Hematology, Central Hospital of Belcolle, 01100 Viterbo, Italy; mariaagnese.fabbri@asl.vt.it (A.F.); enzo.ruggeri@asl.vt.it (E.M.R.); antonella.virtuoso@asl.vt.it (A.V.); carlo.signorelli@asl.vt.it (C.S.); 2Biostatistics Unit, Scientific Directorate, Fondazione Policlinico Universitario A. Gemelli, Istituto di Ricovero e Cura a Carattere Scientifico (IRCCS), 00136 Rome, Italy; diana.giannarelli@policlinicogemelli.it; 3Department of Oncology and Hematology, Thoracic and Interventional Radiology, Central Hospital of Belcolle, 01100 Viterbo, Italy; armando.raso@asl.vt.it (A.R.); fabrizio.chegai@asl.vt.it (F.C.); 4Pathology Unit, Department of Oncology and Hematology, Central Hospital of Belcolle, 01100 Viterbo, Italy; daniele.remotti@asl.vt.it

**Keywords:** COVID-19, mRNA vaccines, booster dose, non-small-cell lung cancer, immune checkpoint inhibitors, first-line therapy, immune-related adverse events, survival

## Abstract

The impact of COVID-19 vaccinations on the safety and efficacy of immune checkpoint inhibitors (ICIs) is still unclear. This research sought to determine whether periodic vaccine boosters could affect the immune-related adverse event (irAE) and survival rates in metastatic non-small cell lung cancer (NSCLC) patients receiving ICIs. The prospective assessment showed no increase in irAE rates or differences in survival outcomes between vaccinated and unvaccinated patients. Among patients with high PD-L1 expression levels who received ICIs alone, vaccine receipt correlated with improved overall survival. Recommended COVID-19 vaccine boosters can be safely administered in patients with advanced NSCLC undergoing immune checkpoint blockade.

## 1. Introduction

The SARS-CoV-2 pandemic has triggered the most severe health emergency in recent history [1]. By the close of 2023, COVID-19 had affected over 770 million individuals and claimed approximately 7 million lives worldwide [2]. Pharmaceutical advancements facilitated the swift creation and authorization of vaccines targeting SARS-CoV-2 spike proteins, utilizing either adenoviral vectors or mRNA technology [3]. The latter has proven to be the most effective measure against the COVID-19 pandemic, curbing infection rates and fostering herd immunity development [4]. Despite the exclusion of frail individuals from mRNA vaccine clinical trials, this group now represents the population of greatest concern regarding COVID-19 risks [5]. Recently, the World Health Organization (WHO) reduced the overall alert status but continues to emphasize SARS-CoV-2 vaccination as a priority for vulnerable populations [6]. Cancer patients undergoing active treatment are of particular interest in this context. While booster doses of mRNA vaccines can mitigate COVID-19 complications, waning immunity and the emergence of new variants have left these patients susceptible to persistent breakthrough SARS-CoV-2 infections [7]. Consequently, expert committees advocate for the regular administration of updated COVID-19 vaccines to cancer patients receiving immunosuppressive therapies [8].

The unprecedented circumstances of the COVID-19 pandemic made it challenging to anticipate the strength of immune responses triggered by mRNA vaccines. Existing guidelines for vaccination, based on experience with cytotoxic therapies, were inadequate in addressing the complexities introduced by immune checkpoint blockade [9]. mRNA vaccines have proven highly effective in stimulating humoral immunity by promoting the production of spike-specific neutralizing antibodies and enhancing adaptive immunity through T cell activation and differentiation [10,11]. Immune checkpoint inhibitors (ICIs) counteract T cell exhaustion by interfering with immunosuppressive signaling between antigen-presenting cells and T cells, thus enhancing anti-tumor immune responses [12]. The immunomodulatory effects of both interventions may result in a complex interplay characterized by mutual enhancement of T cell-mediated responses [13]. These interactions raise concerns about potential clinical consequences, including the intensification of immune-related adverse events (irAEs) and the influence on vaccine efficacy and cancer treatment outcomes [14]. The theoretical synergy has prompted several clinical investigations into patient safety, particularly regarding irAEs [15]. Emerging evidence also suggests that the immunogenicity of COVID-19 mRNA vaccines may enhance the effectiveness of the immune checkpoint blockade across various solid tumors [16]. In addition, recent research has indicated that combining COVID-19 mRNA vaccination with ICIs leads to improved overall survival rates. These enhanced outcomes have been observed across several advanced disease settings, including mixed case series of metastatic cancers [17], melanoma [18], head and neck cancers [19,20], and most notably, non-small cell lung cancer (NSCLC) [20,21]. The available studies were retrospective and based their findings on exposure to the initial two-dose regimen, a vaccine schedule with a reduced immunogenic profile that is no longer recommended [22]. Furthermore, these investigations did not account for the effects of additional booster doses, which are currently the recommended preventive measures against SARS-CoV-2 infection in cancer patients undergoing active treatment [23]. Consequently, we leveraged prospective monitoring from the Vax-On-Third study to examine whether the periodic administration of COVID-19 mRNA vaccines near the initiation of immune checkpoint blockade could influence safety and improve survival outcomes in patients with advanced NSCLC.

## 2. Materials and Methods

### 2.1. Study Design and Eligibility Criteria

The current investigation aimed to evaluate patients with advanced NSCLC who participated in the Vax-On-Third study (EudraCT identifier code: 2021-002611-54). This prospective study adhered to the STROBE guidelines for observational research [24] and received approval from the relevant Ethics Committee (registration code: 1407/CE Lazio1). The same institution also authorized the use of anonymized data for research purposes (protocol number: Oss-R-281; registration code: 855/CE Lazio1). Clinical data were gathered from the National Drug Agency registry, which provides the reimbursement for high-cost drugs through a prospective monitoring of safety and efficacy [25]. All participants gave written informed consent before any procedures were conducted. The primary eligibility criteria for this analysis included a histologically confirmed diagnosis of NSCLC, stage IV disease according to the eighth edition of the AJCC staging system for lung cancer [26], the availability of baseline PD-L1 tumor proportion score (TPS), an Eastern Cooperative Oncology Group Performance Status (ECOG PS) of 0–2, completion of at least two cycles of initial treatment with anti-PD-1 agents (pembrolizumab or cemiplimab) or their combination with platinum-based chemotherapy, absence of genetic mutations suitable for first-line targeted therapies, and a minimum of 12 months of prospective observation following the initiation of treatment. Additionally, patients with disease recurrence following thoracic surgery or definitive radiotherapy were eligible if progression occurred more than 24 weeks after the last perioperative chemotherapy. However, any prior treatment with anti-PD-(L)1 inhibitors was a disqualifying factor for this analysis. Patients with a metastatic involvement of the central nervous system (CNS) could participate if they were asymptomatic or neurologically stable following radiotherapy.

### 2.2. Data Collection and Outcome Assessments

The National Drug Agency was the source of demographic, clinical, pathological, and molecular features, along with treatment outcomes concerning safety, disease response, and survival rates. The agency’s pharmacovigilance database provided insights into the frequency and severity of irAEs [27]. An immunohistochemical evaluation of PD-L1 TPS was conducted using the 22C3 pharmDx anti-PD-L1 antibody and the Dako Omnis platform, following the manufacturer’s guidelines (Agilent Technologies, Inc., Santa Clara, CA, USA). The level of TPS was described as the percentage of at least 100 viable tumor cells showing positive membrane staining for PD-L1 expression [28]. The assessment of PD-L1 TPS required quality control procedures, including internal validation cohorts and confirmation of findings through inter-observer reliability. Before each treatment session, patients received a thorough physical examination and underwent laboratory tests, including standard blood work and assessments of thyroid, renal, hepatobiliary, pancreatic, adrenocortical, pituitary, and muscle functions. The attending physician utilized the Common Criteria for Toxicity (CTC)-AE version 5.0 to define and grade irAEs at each treatment cycle [29]. The evaluation of irAEs was conducted without blinding due to the necessity of reporting to the pharmacovigilance agency. According to the National Drug Agency’s guidelines, a baseline disease assessment was conducted within four weeks of starting treatment. These recommendations also call for an initial disease reassessment between 12 and 16 weeks after treatment commencement and every four to six months thereafter. A blinded radiologist evaluated patients’ records using the Response Evaluation Criteria in Solid Tumors for immunotherapy (iRECIST) [30]. Patients who received any additional mRNA vaccine doses between 60 days before and 30 days after starting anti-PD-1 therapy accounted for the exposed cohort, while those unwilling to receive further boosters constituted the reference cohort. The primary endpoint of this study involved examining the effects of periodic vaccine boosters on irAE, progression-free survival (PFS), and overall survival (OS) rates. The secondary endpoint was to assess these outcomes in relation to PD-L1 TPS levels and subsequent treatment choices (ICIs alone for patients with PD-L1 TPS ≥ 50% or a combination of ICIs and platinum-based chemotherapy for patients with PD-L1 TPS < 50%). PFS was measured from the first administration of ICIs to the date of radiologically ascertained disease progression. OS was calculated from the initial administration of ICIs until the date of certified death, irrespective of the cause. Patients who were free of progressive disease or were still alive by the last follow-up were censored (as of 31 March 2025).

### 2.3. Statistical Analysis

This study applied SPSS version 23.0 (IBM SPSS Statistics for Windows, Armonk, NY, USA) for all statistical assessments and Prism version 9.0 (GraphPad Software Inc., San Diego, CA, USA) for graphical representations. The sample size calculator tailored for observational cohort studies was utilized to determine the number of patients needed for this analysis [31]. The research relied on a superiority design hypothesis. The selected design parameters P0 (the proportion of patients in the unexposed cohort with a PFS duration of at least one year) and P1 (the proportion of patients in the exposed cohort with a PFS duration of at least one year) were set at 0.40 and 0.60, respectively. Given an alpha and beta error probability of 0.05 and 0.80, respectively, a unexposed/exposed ratio of 1.0, and an expected drop rate of 3%, both cohorts required at least 100 cases. Descriptive statistics included a mean with standard deviation (SD) for continuous variables as well as absolute and relative frequencies with interquartile range (IQR) or 95% confidence interval (CI) for categorical variables. We implemented propensity score matching (PSM) to address a potential imbalance of baseline covariates between the cohorts. Propensity scores were estimated through a multivariate logistic regression analysis. This model accounted for the covariates with potential prognostic value, including age, sex, ECOG PS, histological subtype, number of metastatic sites, any specific metastatic involvement (bone, CNS, and/or liver), PD-L1 TPS, body mass index (BMI), smoking habits, previous thoracic radiotherapy, lung immune prognostic index (LIPI) score, treatment regimen, and concomitant medications (corticosteroids, acetaminophen [APAP], antibiotics, and/or proton pump inhibitors [PPIs]). In an effort to ensure adequate numbers of participants and an equal representation in both cohorts, we employed a one-to-one matching approach using the nearest-neighbor algorithm with a 0.2 caliber width. PSM required the application of R software version 4.1.2 and the MatchIt library [32]. Univariate comparisons were performed before and after PSM to demonstrate the balance of prognostic factors using Pearson’s *χ*^2^, Mann–Whitney’s *U* test, or the Kruskal–Wallis test, as appropriate. The balance of covariates between study cohorts was further assessed by calculating the standardized mean difference (SMD), with a value less than 0.1 indicating a well-balanced outcome [33]. The estimation and comparison of PFS and OS relied on the Kaplan–Meier method and the log-rank test, respectively. With regard to primary endpoints related to safety and survival outcomes, univariate and multivariate Cox regression models were applied to calculate hazard ratios (HRs) with 95% CIs and compare rates of immune-related toxicities, disease progression, and mortality. To mitigate alpha inflation from repeated univariate comparisons, the multivariate analysis included vaccine booster exposure as an experimental variable in addition to all covariates used to estimate propensity scores. The subgroup analysis concerning secondary endpoints involved univariate testing only. All tests were two-tailed, and a *p*-value less than 0.05 was considered statistically significant.

## 3. Results

### 3.1. Patient Characteristics

Between 27 November 2021 and 31 March 2024, we successfully enrolled 226 patients who met the eligibility requirements in full. The median (SD) age was 68.9 (11.2) years, and 30.1% were female. All participants had metastatic disease, with the majority showing an ECOG PS of 0 or 1. Based on PD-L1 TPS levels, 93 patients (41.2%) were treated with single-agent anti-PD-1 therapy (either pembrolizumab or cemiplimab), while 133 patients (58.8%) received the combination with platinum-based chemotherapy. Every participant had completed the initial two-dose regimen of tozinameran at least six months prior to their histological diagnosis. The reference group consisted of 114 patients who had not received any booster doses, whereas the exposed group included 112 patients who had received at least one additional vaccination at various times. Following government guidelines, 109 patients (48.2%) received a third dose of tozinameran between 27 September 2021 and 30 June 2022; 20 patients (8.9%) received a fourth dose from 1 July 2022 to 13 September 2022; and 97 patients (42.9%) received the bivalent booster starting 14 September 2022. In univariate comparisons, we observed a significant imbalance in the distribution of ECOG PS scores and previous exposure to chest radiotherapy. PSM was used to achieve a balanced distribution of baseline covariates between the cohorts. After applying PSM, we analyzed a combined group of 102 patients from each cohort. Within the exposed cohort, 13 patients (12.7%) received the booster vaccination after starting ICI therapy, with a median delay of 2 days (ranging from 1 to 6 days). The remaining 89 patients (87.3%) were given the booster vaccination before initiation of treatment, with a median advance of 18 days (ranging from 1 to 50 days). Table 1 depicts the baseline characteristics of PSM-adjusted population that was relevant for all subsequent evaluations. Appendix A provides the baseline characteristics of original unadjusted population.

### 3.2. Safety Analysis

The median duration of anti-PD-1 therapy was 8.0 cycles (IQR 4.0–15.0) in the entire population adjusted for PSM. This statistic was consistent across the reference cohort (8.0 cycles [IQR 3.0–13.5]) and the exposed cohort (8.0 cycles [IQR 4.7–13.0]; *p* = 0.290). Over a median follow-up period of 5.6 months (IQR 2.8–10.3), we observed 72 irAEs, resulting in an overall incidence rate of 35.3% (95% CI 28.2–42.9). Immune-related toxicities involved 64 (31.4%, 95% CI 25.1–38.2) patients, with 8 (3.9%. 95% CI 3.1–4.8) experiencing two simultaneous or sequential adverse events. Immune-related thyroid dysfunctions, skin reactions, colitis, pneumonitis, arthritis, hepatitis, and pancreatitis were the most common toxicities of all grades, with an incidence rate exceeding 2% of cases. Gastrointestinal and pulmonary adverse events were the most frequently reported grade 3 irAEs. No grade 4 immune-related toxicities were observed (Table 2). With the exception of an increased incidence of mild to moderate liver toxicity in the reference subgroup, univariate analysis of any grade irAEs showed no further significant differences between the cohorts (Table 3). Multivariate analysis using comprehensive criteria confirmed that vaccine exposure did not consistently predict the likelihood of irAEs (Appendix A).

### 3.3. Survival Analysis

The median follow-up duration was 22.8 (95% CI 19.2–26.0) months in the PSM-adjusted population. By the cut-off date, 22 (10.8%) patients did not show any disease progression, while 34 (16.7%) patients were censored without experiencing any events relevant to survival. Across the whole population, the median PFS and OS were 7.6 (95% CI 6.5–8.8) months and 13.1 (95% CI 11.5–14.6) months, respectively. The median PFS in the reference and exposed cohorts was 7.5 (95% CI 5.9–9.1) months and 8.2 (95% CI 6.2–10.2) months, respectively (log-rank *p* = 0.408; HR 0.88 [95% CI 0.66–1.18]; Figure 1A). The median OS in the reference and exposed cohorts was 10.5 (95% CI 7.9–13.4) months and 13.8 (95% CI 12.0–15.5) months, respectively (log-rank *p* = 0.170 [HR 0.81 [95% CI 0.59–1.09]; Figure 1B). Univariate and multivariate analyses confirmed the inconsistency of booster vaccine exposure in predicting a different risk of disease progression (Table 4) and mortality (Table 5). Additional univariate comparison involved categorizing patients according to PD-L1 TPS levels. Among the 123 patients (60.3%) with PD-L1 TPS < 50% who received the combination of anti-PD-1 agents and platinum-based chemotherapy, PFS (Figure 2A) and OS (Figure 2B) did not differ significantly between the cohorts. Regarding the subgroup of 81 (39.7%) patients with PD-L1 TPS > 50% who were given anti-PD-1 therapy alone, the comparative assessment confirmed no difference in PFS (Figure 2C) but showed significantly improved OS for patients who received booster vaccination (9.7 [95% CI 8.1–11.2] vs. 18.6 [95% CI 13.5–23.6] months; *p* = 0.034; HR 0.59 [95% CI 0.36–0.96]; Figure 2D).

## 4. Discussion

The simultaneous administration of COVID-19 mRNA vaccines and ICIs has attracted considerable attention due to the hypothetical potential for enhancing immune responses [34]. Initial analyses of the Vax-On-Third study yielded unprecedented results, suggesting increased survival and immune-related thyroid toxicity in patients receiving a third dose of tozinameran during immune checkpoint blockade [35]. However, these preliminary findings were considered tentative due to the limited sample size, heterogeneous case series, and significant variability in vaccination and treatment initiation timing. The present study builds upon these insights to more comprehensively examine the clinical implications of periodic boosters during ICI therapy. To this end, we conducted a longitudinal observation of the Vax-On-Third study, which primarily aimed to assess the safety and efficacy of additional vaccinations during active cancer treatment. Real-world data on clinical outcomes were obtained from the government registry, which monitors drug safety and efficacy prospectively for ICI reimbursement purposes [25]. In an effort to minimize population heterogeneity, we limited our analysis to the upfront treatment of advanced NSCLC, where ICIs are a recognized therapeutic approach [36]. Since the current study was observational using real-world data from current clinical practice, it was essential to achieve an optimal balance of baseline prognostic factors. In accordance with medical research best practices, we implemented a comprehensive weighting system based on relevant clinical, pathological, and pharmacological variables [37]. Furthermore, the most crucial methodological aspect involved precisely defining the interval between vaccination and the start of ICI therapy. Our predetermined time frame aligns with previous influenza vaccination studies [38]. This framework was also consistent with the kinetics of the immune response after the third dose of tozinameran. In this regard, the 90-day period following a vaccine injection is characterized by persistently high anti-spike antibody titers [39] and the emergence of clonal T-cell dominance specific for spike epitopes [40].

Regarding the primary purpose of this research, survival analysis in the general population revealed no differences between the cohorts in terms of PFS or OS. Even though the exposed cohort showed a numerically longer duration for both PFS and OS, the prolongation of survival remained far from reaching statistical significance. These findings do not support the experimental hypothesis of potential superiority associated with periodic exposure to COVID-19 mRNA vaccination and instead indicate a non-inferiority outcome. Likewise, we found no differences in the occurrence rates of irAEs. In this regard, our assessment was thorough in addressing immune-related toxicities during the course of treatment, with an observation period of approximately six months. Although they are less frequent, our safety monitoring may not have detected long-term irAEs as accurately. The latter consideration warrants caution and the need for further surveillance concerning late-onset adverse events. These findings suggest that recommended booster doses of COVID-19 mRNA vaccines, given shortly before or after beginning treatment, do not influence the efficacy and short-term safety of immune checkpoint blockade as upfront therapy for advanced NSCLC. While the results regarding immune-related toxicity rates are largely consistent with the available evidence [41], the survival outcomes of this study do not support previous claims. Earlier published studies on this subject have examined the effects of adenoviral vector vaccines, considered a mixed context of first-line and subsequent therapies, and were retrospective in nature, lacking a precise definition of the temporal relationship between vaccination and ICI therapy duration. Although these issues are relevant and may render even an indirect comparison inconsistent, at least two reasons could underlie the discrepancy with our results. The first explanation is methodological. We believe that prospective data gathering, stringent inclusion criteria, and prognostic balance at baseline mitigated selection bias [42]. This reduces the possibility that unestablished prognostic factors, such as COVID-19 vaccination, might affect survival and makes the analysis of results more reliable. Furthermore, we provided a precise definition of the timing between vaccination and the start of ICI therapy. The time frame was consistent with previous research on influenza vaccination [43] and allowed us to minimize the confounding effects of immortal-time bias [44,45]. An alternative explanation addresses the real extent of immune responses triggered by COVID-19 mRNA vaccines. The enhanced efficacy of immune checkpoint blockade relies on the assumption that administering mRNA vaccines simultaneously boosts T cell functionality, thus enhancing the potential for anticancer immune responses [46]. These vaccines work by delivering genetic instructions to the host for producing specific viral proteins. Upon injection, SARS-CoV-2 mRNA vaccines are detected by various innate sensors, initiating a primary cytokine response involving IFN-γ, IL-2, and IL-4. This response subsequently leads to the production of spike-specific antibodies and the activation and differentiation of T lymphocytes into spike-directed effector cells [11]. While concurrent ICI therapy has been demonstrated to amplify the intensity of the initial cytokine release [47], there is no evidence suggesting that ICI therapy can enhance subsequent T cell-mediated responses. Multiple studies investigating antigen-specific T cell responses have not found significant differences when comparing cancer patients receiving immune checkpoint blockade with healthy volunteers or unvaccinated individuals [48,49]. The most recent data available as preprints in the context of rapidly evolving evidence confirm the latter suggestions [50]. Given that the onset of irAEs may be contingent on the exacerbation of immune responses against self-tissue antigens, the lack of evidence demonstrating an increased risk of immune-related toxicities following COVID-19 mRNA vaccination supports the latter suggestions [51]. In this context, the hypothetical synergy between COVID-19 mRNA vaccines and immune checkpoint blockade is unlikely to result in a more effective anticancer response and improved outcomes.

The secondary aim of this study was to evaluate survival outcomes in relation to PD-L1 expression levels and subsequent treatment choices. Subgroup analysis of patients with low PD-L1 TPS receiving chemotherapy and ICIs confirmed previous results, showing no differences in PFS or OS. However, COVID-19 mRNA vaccine administration was associated with significant OS improvement in high PD-L1 TPS patients undergoing single-agent immune checkpoint blockade. Given the available evidence, the latter finding was not entirely unexpected. Earlier research has shown that SARS-CoV-2 infection can influence PD-1/PD-L1 axis functionality [52]. Patients with severe COVID-19 exhibit upregulation of the PD-1/PD-L1 pathway in several immune cell types, including monocytes, neutrophils, and T cells [53]. These checkpoint molecules may serve as prognostic indicators and potential therapeutic targets [54]. Further studies revealed that COVID-19 mRNA vaccination increases PD-L1 surface expression on peripheral blood granulocytes, monocytes, and intranodal or circulating T helper cells in both healthy and immunocompromised individuals [55,56]. Recent preclinical data have shown that intratumoral injections of tozinameran induce abundant tumor-infiltrating lymphocytes with enhanced local levels of proinflammatory markers and significantly increased expression of PD-L1 on tumor-associated immune cells. These effects would result in changing the tumor immune microenvironment to a state more favorable for the therapeutic efficacy of immune checkpoint blockade [57]. Assuming that mRNA vaccines targeting SARS-CoV-2 would similarly increase the PD-L1 TPS, more recent studies found that patients with advanced NSCLC were more likely to have elevated expression of this checkpoint molecule if they were vaccinated less than 100 days before diagnostic biopsy [58]. Notably, better OS rates were observed in recipients of COVID-19 mRNA vaccines within 100 days of starting ICI therapy than in unvaccinated patients [59]. Although the mechanisms underlying the modulation of the PD-1/PD-1 axis by these vaccines remain unclear, our subgroup analysis seems consistent with the latter findings. The current study’s results suggest that the survival benefit is confined to NSCLC patients with high PD-L1 TPS who received vaccination near the start of immune checkpoint blockade. However, caution is warranted due to significant differences in study conditions. Unlike the study by Grippin et al., where unvaccinated patients served as the reference group, all participants in our research had received an initial two-dose tozinameran series at least six months before diagnosis. Additionally, most recipients in the former study likely received initial COVID-19 mRNA vaccination priming, resulting in an immune response profile distinct from that induced by booster doses in our study cohort.

Of course, this study acknowledges several limitations, which may extend beyond the following issues. Firstly, despite the prospective nature of this research, the analysis relied on real-world data from consecutively enrolled patients without baseline stratification of prognostic factors. This methodology inherently carries confounding and selection bias, even with the use of strict inclusion criteria and prognostic balancing through an extensive PSM. Secondly, our assessments of treatment failure relied on radiological examinations that were blinded but not independent. Lack of external review of the disease response may lead to an overestimation of the efficacy of immune checkpoint blockade in terms of PFS. Thirdly, we considered patients who had received less than two cycles of treatment to be ineligible for safety and efficacy analysis. These patients generally represent a frail population with a poor prognosis [60]. Although the subgroup is small, their exclusion introduces a selection bias with a potential impact on the assessment of survival outcomes. The emerging evidence that a poor frailty score might predict an increased risk of irAE implies an additional limitation that affects the generalizability of our findings [61]. Fourthly, the current research relies on an accurate definition of the temporal relationship between vaccination and initiation of anti-PD-1 therapy. This experimental design minimizes the effects of immortal-time bias but implies that our findings may not be applicable to other vaccination schedules over the course of treatment. In addition, the interval between vaccination and the start of ICI therapy, which was noticeable among patients vaccinated in advance, may represent a residual confounding. Given the potential differences in immunogenicity among different vaccine booster types, a stratified analysis addressing this heterogeneity would be valuable. However, the numerical distribution of patients makes this subgroup statistic underpowered and therefore unreliable. Lastly, we must consider the risk of alpha inflation due to multiple univariate comparisons [62]. Although our multivariate analysis was thorough, incorporating all potential prognostic factors to reduce the risk of false-negative results in an unprecedented experimental context, the potential for false-positive results remains inherent in this methodology.

## 5. Conclusions

International consensus supports a regular mRNA vaccination as the most effective measure to prevent severe COVID-19 in cancer patients undergoing immunosuppressive treatments. This study provides initial evidence that recommended vaccine boosters are safe and do not influence expected survival rates in advanced NSCLC patients receiving ICIs, with or without chemotherapy. Consistent with the still preliminary data, the subgroup analysis even suggests a potential survival benefit for vaccinated patients with high PD-L1 expression who are treated exclusively with anti-PD-1 agents. However, due to study limitations and the lack of experimental evidence for reliable comparisons, these findings must be considered exploratory. Their value as hypothesis-generating results warrants prospective validation in independent series and, ideally, through randomized controlled trials with correlative biomarker analyses. Our findings are consistent with the recommendations of relevant guidelines for vaccination of cancer patients on active treatment and do not suggest a personalized immunization strategy based on different safety and efficacy profiles [63]. In agreement with the latest advances and recommendations, the primary rationale for advocating mRNA vaccination in advanced NSCLC patients eligible for immune checkpoint blockade is the prevention of severe COVID-19 [64].

## Figures and Tables

**Figure 1 cancers-17-01948-f001:**
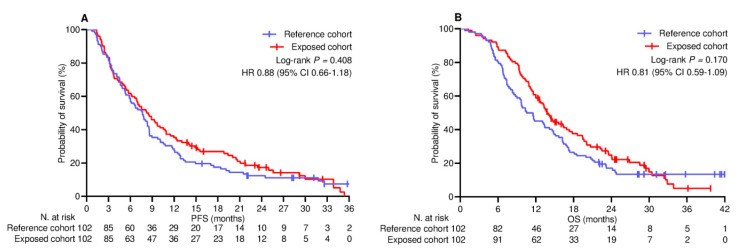
Univariate comparison of survival in PSM−adjusted population (N = 204). (**A**) Progression-free survival; (**B**) Overall survival. PSM, propensity score matching; HR, hazard ratio; CI, confidence interval.

**Figure 2 cancers-17-01948-f002:**
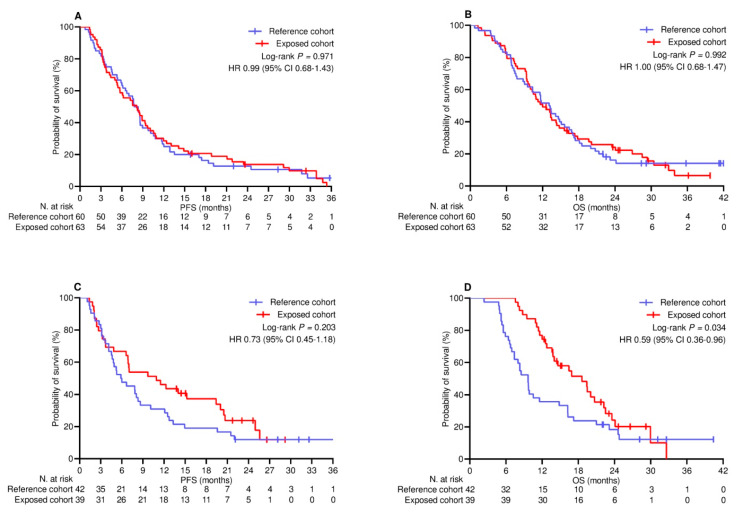
Univariate comparison of survival by PD─L1 TPS. (**A**) PFS in patients with PD-L1 < 50% receiving ICIs and platinum-based chemotherapy; (**B**) OS in patients with PD-L1 < 50% receiving ICIs and platinum-based chemotherapy; (**C**) PFS in patients with PD-L1 ≥ 50% receiving ICIs alone; (**D**) OS in patients with PD-L1 ≥ 50% receiving ICIs alone. PD-L1 TPS, programmed cell death ligand-1 tumor proportion score; PFS, progression-free survival; OS, overall survival; HR, hazard ratio; CI, confidence interval; ICIs, immune checkpoint inhibitors.

**Table 1 cancers-17-01948-t001:** Patient characteristics of PSM-adjusted population.

Variable	All Patients (N = 204)	Reference Cohort (N = 102)	Exposed Cohort (N = 102)	*p* Value	SMD
Age - mean (SD), years - ≥70 years	68.8 (8.4) 110 (53.9%)	69.0 (8.1) 58 (56.9%)	68.9 (8.7) 52 (51.0%)	0.740 0.483	- 0.058
Sex - female - male	67 (32.8%) 137 (67.2%)	32 (31.4%) 70 (68.6%)	35 (34.3%) 67 (65.7%)	0.766	0.029
ECOG PS - 0 - 1 - 2	49 (24.0%) 123 (60.3%) 32 (15.7%)	31 (30.4%) 54 (52.9%) 17 (16.7%)	18 (17.7%) 69 (67.6%) 15 (14.7%)	0.067	<0.001
Histology - non-squamous - squamous	147 (72.1%) 57 (27.9%)	78 (76.5%) 24 (23.5%)	69 (67.6%) 33 (32.4%)	0.212	<0.001
Metastatic sites - ≤2 - >2	115 (56.4%) 89 (43.6%)	55 (53.9%) 47 (46.1%)	60 (58.8%) 42 (41.2%)	0.572	0.049
Bone metastasis	46 (22.5%)	25 (24.5%)	21 (20.6%)	0.616	0.039
CNS metastasis	47 (23.0%)	19 (18.6%)	28 (27.5%)	0.1830	<0.001
Liver metastasis	18 (8.8%)	9 (8.8%)	9 (8.8%)	1	0.001
PD-L1 TPS - <1% - ≥1% and ≤49% - ≥50%	64 (31.4%) 59 (28.9%) 81 (39.7%)	33 (32.4%) 27 (26.5%) 42 (41.2%)	31 (30.4%) 32 (31.4%) 39 (38.2%)	0.742	0.039
BMI - mean (SD), kg/m^2^ - ≥25 kg/m^2^	25.8 (4.4) 89 (43.6%)	25.9 (4.9) 49 (48.0%)	25.7 (3.9) 40 (39.2%)	0.985 0.259	- 0.088
Smoking habits - never - current or former	17 (8.3%) 187 (91.7%)	8 (7.8%) 94 (92.2%)	9 (8.8%) 93 (91.2%)	1	0.009
Previous thoracic RT	35 (17.2%)	12 (11.8%)	23 (22.5%)	0.062	<0.001
LIPI category - 0 - 1 - 2	78 (38.2%) 84 (41.2%) 42 (20.6%)	42 (41.2%) 35 (34.3%) 25 (24.5%)	36 (35.3%) 49 (48.0%) 17 (16.7%)	0.115	0.019
Upfront therapy - only ICIs - pemetrexed-based - paclitaxel-based	81 (39.7%) 86 (42.2%) 37 (18.1%)	42 (41.2%) 46 (45.1%) 14 (13.7%)	39 (38.2%) 40 (39.2%) 23 (22.5%)	0.257	<0.001
Corticosteroid therapy ^a^	88 (43.1%)	39 (38.2%)	49 (48.0%)	0.203	<0.001
APAP ^b^	79 (38.7%)	35 (34.3%)	44 (43.1%)	0.250	<0.0001
Systemic antimicrobial therapy ^c^	44 (21.6%)	23 (22.5%)	21 (20.6%)	0.865	0.019
PPI ^d^	67 (32.8%)	35 (34.3%)	32 (31.4%)	0.766	0.029

PSM, propensity score matching; SMD, standardized mean difference; SD, standard deviation; ECOG PS, Eastern Cooperative Oncology Group Performance Status; CNS, central nervous system; PD-L1 TPS, programmed cell death ligand-1 tumor proportion score; BMI, body mass index; RT, radiotherapy; LIPI, lung immune prognostic index; APAP, acetaminophen; PPI, proton pump inhibitors. ^a^ exposure to high dose corticosteroid drugs (prednisone equivalent ≥ 10 mg daily for at least 5 days) within the 30 days prior to the start of treatment (not including premedication for chemotherapy); ^b^ exposure to therapeutic dose of APAP (at least 1000 mg per day for more than 24 h) during the 30 days prior to the start of treatment; ^c^ exposure to therapeutic dose of any systemic antibiotics in the 30 days prior to the start of treatment; ^d^ exposure to any PPI dose at the start of treatment.

**Table 2 cancers-17-01948-t002:** Immune-related adverse events in PSM-adjusted population (N = 204).

irAE Type	All Grades, No. of Patients (%)	Grade 1–2, No. of Patients (%)	Grade 3–4, No. of Patients (%)	Median Time to Onset, Weeks (IQR)
All types	72 (35.3%)	58 (28.4%)	14 (6.9%)	-
Thyroid dysfunction - Hypothyroidism - Hyperthyroidism	13 (6.4%) 2 (1.0%)	12 (5.9%) 2 (1.0%)	1 (0.5%) -	10.2 (6.7–22.7) 7.1 (5.9–11.4)
Dermatologic	11 (5.4%)	10 (4.9%)	1 (0.5%)	6.9 (3.2–17.4)
Colitis	8 (3.9%)	6 (2.9%)	2 (1.0%)	6.6 (2.9–28.1)
Pneumonitis	7 (3.4%)	4 (1.9%)	3 (1.5%)	12.3 (7.5–20.8)
Hepatitis	5 (2.4%)	4 (1.9%)	1 (0.5%)	4.9 (2.2–22.3)
Arthritis	6 (2.9%)	6 (2.4%)	-	37.5 (16.2–49.2)
Pancreatitis	5 (2.4%)	4 (1.9%)	1 (0.5%)	9.9 (5.6–25.1)
Myositis	3 (1.5%)	3 (1.4%)	-	13.2 (6.9–20.6)
Nephritis	2 (1.0%)	1 (0.5%)	1 (0.5%)	11.9 (3.4–20.3)
Diabetes	2 (1.0%)	2 (1.0%)	-	10.8 (6.2–16.5)
Hypophysitis	2 (1.0%)	1 (0.5%)	1 (0.5%)	20.6 (7.6–37.1)
Vasculitis	2 (1.0%)	1 (0.5%)	1 (0.5%)	5.0 (3.2–6.9)
Adrenal dysfunction	1 (0.5%)	1 (0.5%)	-	12.8 (9.6–30.4)
Peripheral sensory neuropathy	1 (0.5%)	-	1 (0.5%)	6.7 (3.1–32.1)
Uveitis	1 (0.5%)	-	1 (0.5%)	9.1 (7.0–37.6)
Myocarditis	1 (0.5%)	1 (0.5%)	-	4.9 (4.3–11.4)

PSM, propensity score matching; irAE, immune-related adverse event; IQR, interquartile range.

**Table 3 cancers-17-01948-t003:** Immune-related adverse events by vaccine exposure cohorts in PSM-adjusted population (N = 204).

irAE Type	All Grades, No. of Patients (%)			Grade 1–2, No. of Patients (%)			Grade 3, No. of Patients (%)		
	Reference Cohort (N = 102)	Exposed Cohort (N = 102)	*p* Value	Reference Cohort (N = 102)	Exposed Cohort (N = 102)	*p* Value	Reference Cohort (N = 102)	Exposed Cohort (N = 102)	*p* Value
All types	37 (36.3%)	35 (34.3%)	0.769	33 (32.4%)	25 (24.5%)	0.214	4 (3.9%)	10 (9.8%)	0.096
Dermatologic	4 (3.9%)	7 (6.8%)	0.369	4 (3.9%)	6 (5.9%)	0.516	-	1 (0.5%)	0.316
Thyroid dysfunction	7 (6.8%)	8 (7.8%)	0.788	7 (6.8%)	7 (6.8%)	1	-	1 (0.5%)	0.316
Colitis	3 (2.9%)	5 (4.9%)	0.470	3 (2.9%)	3 (2.9%)	1	-	2 (1.0%)	0.155
Pneumonitis	4 (3.9%)	3 (2.9%)	0.700	2 (1.0%)	2 (1.0%)	1	2 (1.0%)	1 (0.5%)	0.560
Hepatitis	4 (3.9%)	1 (0.5%)	0.174	4 (3.9%)	-	0.043	-	1 (0.5%)	0.316
Arthritis	4 (3.9%)	2 (1.0%)	0.407	4 (3.9%)	2 (1.0%)	0.407	-	-	-
Pancreatitis	2 (1.0%)	3 (2.9%)	0.650	2 (1.0%)	2 (1.0%)	1	-	1 (0.5%)	0.316
Myositis	2 (1.0%)	1 (0.5%)	0.560	2 (1.0%)	1 (0.5%)	0.560	-	-	-
Nephritis	2 (1.0%)	-	0.155	1 (0.5%)	-	0.316	1 (0.5%)	-	0.316
Diabetes	1 (0.5%)	1 (0.5%)	1	1 (0.5%)	1 (0.5%)	1	-	-	-
Hypophysitis	-	2 (1.0%)	0.155	-	1 (0.5%)	0.316	-	1 (0.5%)	0.316
Vasculitis	1 (0.5%)	1 (0.5%)	1	1 (0.5%)	-	0.316	-	1 (0.5%)	0.316
Adrenal dysfunction	1 (0.5%)	-	0.316	1 (0.5%)	-	0.316	-	-	-
Peripheral sensory neuropathy	-	1 (0.5%)	0.316	-	-	-	-	1 (0.5%)	0.316
Uveitis	1 (0.5%)	-	0.316	-	-	-	1 (0.5%)	-	0.316
Myocarditis	1 (0.5%)	-	0.316	1 (0.5%)	-	0.316	-	-	-

irAE, immune-related adverse event; IQR, interquartile range.

**Table 4 cancers-17-01948-t004:** Analysis of progression-free survival in PSM-adjusted population (N = 204).

Covariate	Median PFS, Months (95% CI)	Univariate Analysis		Multivariate Analysis	
		HR (95% CI)	*p* Value	HR (95% CI)	*p* Value
Age - <70 years (N = 94) - ≥70 years (N = 110)	7.6 (6.0–9.3) 7.7 (6.3–9.1)	1.00 1.05 (0.78–1.40)	- 0.739	1.00 0.94 (0.66–1.33)	- 0.748
Sex - female (N = 67) - male (N = 137)	7.7 (3.6–11.9) 7.8 (6.7–8.9)	1.00 0.78 (0.57–1.06)	- 0.123	1.00 0.84 (0.58–1.22)	- 0.373
ECOG PS - 0 (N = 49) - 1 (N = 123) - 2 (N = 32)	11.9 (7.9–15.9) 7.7 (6.5–8.9) 3.7 (2.2–5.2)	1.00 1.87 (1.28–2.72) 2.81 (1.73–4.55)	- 0.001 <0.001	1.00 1.14 (0.71–1.83) 1.77 (0.99–3.15)	- 0.584 0.052
Histology - non-squamous (N = 147) - squamous (N = 57)	7.6 (6.3–8.9) 8.5 (7.0–10.0)	1.00 0.86 (0.62–1.20)	- 0.394	1.00 0.93 (0.55–1.59)	- 0.809
Metastatic sites - ≤2 (N = 115) - >2 (N = 89)	8.5 (7.5–9.5) 7.0 (4.8–9.1)	1.00 1.22 (0.91–1.64)	- 0.172	1.00 1.11 (0.68–1.82)	- 0.670
Bone metastasis - no (N = 158) - any (N = 46)	8.4 (7.4–9.5) 5.2 (2.6–7.7)	1.00 1.75 (1.25–2.46)	- 0.001	1.00 1.04 (0.65–1.66)	- 0.854
CNS metastasis - no (N = 157) - any (N = 47)	7.8 (6.4–9.2) 7.7 (4.9–10.5)	1.00 0.83 (0.59–1.18)	- 0.324	1.00 0.73 (0.43–1.24)	- 0.256
Liver metastasis - no (N = 186) - any (N = 18)	8.0 (7.0–9.1) 6.2 (2.7–9.6)	1.00 1.06 (0.62–1.81)	- 0.806	1.00 1.01 (0.55–1.82)	- 0.990
PD-L1 TPS - <1% (N = 64) - ≥1% and ≤49% (N = 59) - ≥50% (N = 81)	6.7 (4.6–8.7) 8.9 (6.8–11.1) 7.0 (5.3–8.6)	1.00 0.74 (0.51–1.07) 0.75 (0.53–1.07)	- 0.117 0.122	1.00 0.82 (0.53–1.25) 0.75 (0.16–3.32)	- 0.365 0.696
BMI - <25 kg/m^2^ (N = 115) - ≥25 kg/m^2^ (N = 89)	7.0 (5.6–8.3) 8.6 (7.9–9.2)	1.00 0.94 (0.70–1.26)	- 0.688	1.00 1.04 (0.74–1.47)	- 0.801
Smoking habits - never (N = 17) - current or former (N = 187)	4.9 (2.4–7.3) 8.1 (7.0–9.1)	1.00 0.67 (0.40–1.11)	- 0.121	1.00 0.77 (0.42–1.41)	- 0.400
Previous chest radiotherapy - no (N = 169) - yes (N = 35)	7.8 (6.4–9.2) 7.6 (5.2–10.0)	1.00 0.90 (0.61–1.32)	- 0.597	1.00 0.93 (0.60–1.44)	- 0.756
LIPI category - 0 (N = 78) - 1 (N = 84) - 2 (N = 42)	13.5 (7.7–19.3) 6.9 (5.4–8.3) 3.0 (2.5–3.4)	1.00 2.70 (1.90–3.85) 8.52 (5.52–13.16)	- <0.001 <0.001	1.002.79 (1.87–4.17) 9.25 (5.63–15.20)	- <0.001 <0.001
Upfront therapy - only pembrolizumab (N = 81) - pemetrexed-based (N = 86) - paclitaxel-based (N = 37)	7.0 (5.3–8.6) 8.5 (5.5–9.5) 8.5 (8.0–9.0)	1.00 1.18 (0.86–1.64) 0.86 (0.56–1.32)	- 0.293 0.513	1.00 1.08 (0.24–4.75) 0.66 (0.15–2.90)	- 0.912 0.590
Corticosteroid therapy ^a^ - no (N = 116) - yes (N = 88)	10.1 (7.6–12.5) 4.2 (2.8–5.6)	1.00 2.05 (1.52–2.76)	- <0.001	1.00 1.63 (1.15–2.31)	- 0.006
APAP ^b^ - no (N = 125) - yes (N = 79)	8.6 (7.2–9.9) 6.1 (4.4–7.7)	1.00 1.22 (0.90–1.64)	- 0.185	1.00 1.10 (0.76–1.60)	- 0.584
Systemic antimicrobial therapy ^c^ - no (N = 160) - yes (N = 44)	8.6 (7.0–10.2) 4.1 (2.5–5.7)	1.00 2.27 (1.59–3.23)	- <0.001	1.00 1.57 (1.02–2.43)	- 0.040
PPI ^d^ - no (N = 137) - yes (N = 67)	8.3 (7.0–9.5) 7.0 (5.7–8.2)	1.00 1.28 (0.94–1.75)	- 0.109	1.00 1.37 (0.94–1.99)	- 0.093
Vaccine exposure - no (N = 102) - yes (N = 102)	7.6 (5.8–9.4) 8.2 (6.2–10.2)	1.00 0.88 (0.66–1.18)	- 0.410	1.00 0.92 (0.65–1.29)	- 0.637

PFS, progression-free survival; CI, confidence interval; HR, hazard ratio; ECOG PS, Eastern Cooperative Oncology Group Performance Status; PD-L1 TPS, programmed cell death ligand-1 tumor proportion score; BMI, body mass index; LIPI, lung immune prognostic index; APAP, acetaminophen; PPI, proton pump inhibitors. ^a^ exposure to high dose corticosteroid drugs (prednisone equivalent ≥ 10 mg daily for at least 5 days) within the 30 days prior to the start of treatment (not including premedication for chemotherapy); ^b^ exposure to therapeutic dose of APAP (at least 1000 mg per day for more than 24 h) during the 30 days prior to the start of treatment; ^c^ exposure to therapeutic dose of any systemic antibiotics in the 30 days prior to the start of treatment; ^d^ exposure to any PPI dose at the start of treatment.

**Table 5 cancers-17-01948-t005:** Analysis of overall survival in PSM-adjusted population (N = 204).

Covariate	Median OS, Months (95% CI)	Univariate Analysis		Multivariate Analysis	
		HR (95% CI)	*p* Value	HR (95% CI)	*p* Value
Age - <70 years (N = 94) - ≥70 years (N = 110)	12.7 (9.4–16.0) 13.0 (11.3–14.6)	1.00 1.05 (0.77–1.42)	- 0.740	1.00 0.86 (0.59–1.26)	- 0.451
Sex - female (N = 67) - male (N = 137)	11.6 (8.7–14.4) 13.3 (11.2–15.3)	1.00 0.80 (0.58–1.10)	- 0.172	1.00 0.80 (0.55–1.18)	- 0.274
ECOG PS - 0 (N = 49) - 1 (N = 12) - 2 (N = 32)	16.2 (11.7–20.7) 13.0 (11.5–14.6) 9.5 (6.6–12.3)	1.00 1.70 (1.15–2.51) 2.43 (1.49–3.97)	- 0.007 <0.001	1.00 1.19 (0.74–1.93) 1.75 (0.98–3.14)	- 0.459 0.059
Histology - non-squamous (N = 147) - squamous (N = 57)	13.3 (10.9–15.7) 12.7 (11.0–14.4)	1.00 0.94 (0.67–1.32)	- 0.741	1.00 0.87 (0.50–1.50)	- 0.624
Metastatic sites - ≤2 (N = 115) - >2 (N = 89)	13.3 (11.2–15.4) 11.7 (8.4–15.1)	1.00 0.98 (0.72–1.33)	- 0.910	1.00 0.84 (0.50–1.41)	- 0.520
Bone metastasis - no (N = 158) - any (N = 46)	13.2 (11.2–15.1) 10.3 (4.9–15.6)	1.00 1.40 (0.99–1.99)	- 0.057	1.00 0.85 (0.52–1.38)	- 0.515
CNS metastasis - no (N = 157) - any (N = 47)	13.0 (11.2–14.1) 13.1 (9.5–16.8)	1.00 0.69 (0.48–1.01)	- 0.061	1.00 0.65 (0.37–1.12)	- 0.123
Liver metastasis - no (N = 186) - any (N = 18)	13.1 (11.3–14.9) 10.2 (4.2–16.2)	1.00 1.16 (0.67–2.01)	- 0.590	1.00 1.28 (0.67–2.45)	- 0.444
PD-L1 TPS - <1% (N = 64) - ≥1% and ≤49% (N = 59) - ≥50% (N = 81)	10.3 (8.0–12.6) 13.5 (11.4–15.5) 13.8 (9.9–17.6)	1.00 0.73 (0.50–1.08) 0.77 (0.54–1.11)	- 0.119 0.171	1.00 0.70 (0.45–1.10) 0.32 (0.04–2.27)	- 0.129 0.259
BMI - <25 kg/m^2^ (N = 115) - ≥25 kg/m^2^ (N = 89)	13.3 (11.4–15.1) 12.0 (9.1–14.8)	1.00 1.01 (0.74–1.36)	- 0.960	1.00 1.13 (0.80–1.60)	- 0.463
Smoking habits - never (N = 17) - current or former (N = 187)	14.1 (8.9–19.3) 12.7 (11.2–14.3)	1.00 1.07 (0.61–1.85)	- 0.805	1.00 1.60 (0.82–3.08)	- 0.161
Previous chest radiotherapy - no (N = 169) - yes (N = 35)	12.6 (10.8–14.4) 13.4 (10.3–16.5)	1.00 0.99 (0.66–1.41)	- 0.966	1.00 1.03 (0.67–1.60)	- 0.863
LIPI category - 0 (N = 78) - 1 (N = 84) - 2 (N = 42)	20.7 (17.3–24.03) 11.6 (9.9–13.3) 6.4 (4.9–7.8)	1.00 2.47 (1.71–3.56) 6.85 (4.41–10.6)	- <0.001 <0.001	1.00 2.91 (1.90–4.43) 7.58 (4.57–12.56)	- <0.001 <0.001
Upfront therapy - only pembrolizumab (N = 81) - pemetrexed-based (N = 86) - paclitaxel-based (N = 37)	13.6 (10.5–16.6) 11.7 (8.4–15.0) 12.6 (10.5–14.6)	1.00 1.08 (0.77–1.50) 0.96 (0.62–1.48)	- 0.6490.861	1.00 0.55 (0.08–3.71) 0.43 (0.06–2.85)	- 0.542 0.389
Corticosteroid therapy ^a^ - no (N = 116) - yes (N = 88)	16.2 (13.2–19.2) 10.2 (8.5–12.0)	1.00 2.05 (1.51–2.78)	- <0.001	1.00 1.64 (1.15–2.33)	- 0.006
APAP ^b^ - no (N = 125) - yes (N = 79)	13.3 (10.7–15.8) 12.0 (9.2–14.7)	1.00 1.19 (0.88–1.62)	- 0.252	1.00 1.10 (0.75–1.60)	- 0.618
Systemic antimicrobial therapy ^c^ - no (N = 160) - yes (N = 44)	14.1 (11.9–16.3) 9.6 (7.7–11.5)	1.00 1.99 (1.39–2.85)	- <0.001	1.00 1.36 (0.89–2.07)	- 0.147
PPI ^d^ - no (N = 137) - yes (N = 67)	13.8 (11.3–16.2) 11.6 (9.2–13.9)	1.00 1.37 (1.00–1.89)	- 0.047	1.00 1.31 (0.90–1.92)	- 0.153
Vaccine exposure - no (N = 102) - yes (N = 102)	10.5 (7.7–13.3) 13.8 (12.0–15.5)	1.00 0.81 (0.59–1.09)	- 0.171	1.00 0.69 (0.48–1.01)	- 0.056

OS, overall survival; CI, confidence interval; HR, hazard ratio; ECOG PS, Eastern Cooperative Oncology Group Performance Status; PD-L1 TPS, programmed cell death ligand-1 tumor proportion score; BMI, body mass index; LIPI, lung immune prognostic index; APAP, acetaminophen; PPI, proton pump inhibitors. ^a^ exposure to high dose corticosteroid drugs (prednisone equivalent ≥ 10 mg daily for at least 5 days) within the 30 days prior to the start of treatment (not including premedication for chemotherapy); ^b^ exposure to therapeutic dose of APAP (at least 1000 mg per day for more than 24 h) during the 30 days prior to the start of treatment; ^c^ exposure to therapeutic dose of any systemic antibiotics in the 30 days prior to the start of treatment; ^d^ exposure to any PPI dose at the start of treatment.

## Data Availability

The datasets generated and analyzed during the current study are available from the corresponding author on reasonable request.

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
