# Peer review of "Periodic Boosters of COVID-19 Vaccines Do Not Affect the Safety and Efficacy of Immune Checkpoint Inhibitors for Advanced Non-Small Cell Lung Cancer: A Longitudinal Analysis of the Vax-On-Third Study"

_cancers, 2025, doi:10.3390/cancers17121948_

Round 1
Reviewer 1 Report
Comments and Suggestions for Authors
This prospective cohort study investigates the safety and efficacy of COVID-19 vaccine boosters in combination with immune checkpoint inhibitors (ICIs) for advanced non-small cell lung cancer (NSCLC), addressing a critical clinical question with high relevance during the ongoing pandemic. The use of propensity score matching (PSM) to balance baseline characteristics and the focus on booster doses (rather than primary vaccination) are notable strengths. The finding of improved overall survival (OS) in high PD-L1 expressing patients adds an important nuance to the field. However, several methodological and interpretative considerations warrant attention to strengthen the manuscript.
Reviewer Comments and Recommendations
- Study Design and Methodology
- Vaccination Window DefinitionThe study defines the exposure window as "60 days before to 30 days after initiating ICIs," referencing influenza vaccine studies. While this temporal framework is reasonable, the manuscript should explicitly justify how this aligns with the immunokinetic profile of mRNA vaccines (e.g., peak antibody/t-cell responses). Clarification on whether this window was pre-specified in the study protocol would enhance transparency.
- PD-L1 Assay StandardizationThe use of the 22C3 pharmDx assay on the Dako platform is appropriate, but the manuscript lacks details on inter-batch variability or independent pathological review. Including quality control measures (e.g., internal validation cohorts, inter-observer reliability) would strengthen confidence in PD-L1 stratification.
- irAE Monitoring ProtocolThe assessment of immune-related adverse events (irAEs) relies on clinical records, but the frequency of monitoring (e.g., routine blood work, thyroid function tests) and the blinding of assessors are not specified. Standardized monitoring protocols (e.g., monthly labs) should be described to minimize detection bias.
- Results and Statistical Analysis
- Subgroup Analysis Power and Multiple TestingThe significant OS benefit in the high PD-L1 subgroup (TPS ≥50%, ICIs monotherapy) is intriguing but requires context. Was this subgroup pre-specified in the study hypothesis? Given the exploratory nature of this analysis, adjusting for multiple comparisons (e.g., Bonferroni correction) or reporting a false discovery rate would mitigate Type I error concerns. Additionally, clarify the sample size after PSM in this subgroup (e.g., n=? per arm).
- Exclusion of Short-Term ICI UsersPatients receiving <2 cycles of ICIs were excluded, which may introduce selection bias (these patients often have poor prognosis). Discuss how this exclusion might affect generalizability, particularly regarding safety outcomes (e.g., irAEs in frail populations).
- Heterogeneity of Booster TypesThe exposed cohort includes patients receiving 3rd dose, 4th dose, or bivalent boosters. Given potential differences in immunogenicity between booster types, stratified analyses (e.g., 3rd vs. 4th dose) or a sensitivity analysis addressing this heterogeneity would be valuable. If underpowered, this limitation should be explicitly acknowledged.
- Discussion and Interpretation
- Mechanistic Hypotheses for PD-L1 Subgroup EffectThe proposed mechanism linking mRNA vaccination to PD-L1 upregulation and enhanced ICI response is plausible but requires stronger mechanistic support. Cite preclinical studies showing mRNA vaccine-induced PD-L1 expression in antigen-presenting cells or tumor cells (e.g., studies on TLR activation or IFN-γ signaling).
- Comparison with Prior StudiesThe manuscript notes discrepancies with retrospective studies (e.g., Grippin et al.), which may relate to differences in vaccination timing (booster vs. primary series) or baseline immunity (all patients here had prior 2-dose priming). Elaborate on these contextual differences to provide a more nuanced discussion.
- Cautiousness in ConclusionWhile the PD-L1 subgroup finding is hypothesis-generating, it should be explicitly framed as exploratory rather than definitive. Emphasize the need for prospective validation in randomized controlled trials (RCTs), ideally with correlative biomarker analyses.
- Other Recommendations
- Data TransparencyInclude a table of standardized differences post-PSM to demonstrate balance of covariates (e.g., ECOG PS, metastatic sites) between cohorts, as recommended in observational research reporting guidelines (e.g., STROBE).
- Update Reference CitationsReplace preprint citations (e.g., Ravera et al., 2024) with peer-reviewed publications where possible, or justify the use of preprints in the context of rapidly evolving evidence.
- Guideline AlignmentDiscuss how these findings inform current vaccination guidelines for cancer patients (e.g., ASCO 2024, NCCN 2024), highlighting the safety profile and potential subgroup benefits to support personalized vaccination strategies.
Author Response
Response to Reviewer 1 Comments
Comments and Suggestions for Authors
This prospective cohort study investigates the safety and efficacy of COVID-19 vaccine boosters in combination with immune checkpoint inhibitors (ICIs) for advanced non-small cell lung cancer (NSCLC), addressing a critical clinical question with high relevance during the ongoing pandemic. The use of propensity score matching (PSM) to balance baseline characteristics and the focus on booster doses (rather than primary vaccination) are notable strengths. The finding of improved overall survival (OS) in high PD-L1 expressing patients adds an important nuance to the field. However, several methodological and interpretative considerations warrant attention to strengthen the manuscript.
Reviewer Comments and Recommendations
Study Design and Methodology
Vaccination Window DefinitionThe study defines the exposure window as "60 days before to 30 days after initiating ICIs," referencing influenza vaccine studies. While this temporal framework is reasonable, the manuscript should explicitly justify how this aligns with the immunokinetic profile of mRNA vaccines (e.g., peak antibody/t-cell responses). Clarification on whether this window was pre-specified in the study protocol would enhance transparency.
- We greatly appreciate these insightful comments. The exposure window of “60 days before and 30 days after the start of ICIs” relied primarily on evidence from studies that investigated interactions between influenza vaccination and immune checkpoint blockade. However, this time frame was also consistent with the kinetics of the immune response after the third dose of tozinameran. In this regard, the 90-day period following vaccine injection is characterized by persistently high levels of anti-spike antibody titers [Kim HH, et al. Time-course analysis of antibody and cytokine response after the third SARS-CoV-2 vaccine dose. Vaccine X, 20:100565] and the development of clonal T-cell dominance specific for spike epitopes [Aoki H, et al. CD8+ T cell memory induced by successive SARS-CoV-2 mRNA vaccinations is characterized by shifts in clonal dominance. Cell Rep. 2024 Mar 26;43(3):113887]. According to the Reviewer's suggestion, we have added these comments to the “Discussion” section with appropriate references.
PD-L1 Assay StandardizationThe use of the 22C3 pharmDx assay on the Dako platform is appropriate, but the manuscript lacks details on inter-batch variability or independent pathological review. Including quality control measures (e.g., internal validation cohorts, inter-observer reliability) would strengthen confidence in PD-L1 stratification.
- The analysis of PD-L1 TPS also involved quality control measures, including internal validation cohorts and confirmation of results through inter-observer reliability. We have added this statement to subsection “2.2 Data Collection and Outcome Evaluations.”
irAE Monitoring ProtocolThe assessment of immune-related adverse events (irAEs) relies on clinical records, but the frequency of monitoring (e.g., routine blood work, thyroid function tests) and the blinding of assessors are not specified. Standardized monitoring protocols (e.g., monthly labs) should be described to minimize detection bias.
- Before each treatment session, patients received a thorough physical examination and underwent laboratory tests, including standard blood work and assessments of thyroid, renal, hepatobiliary, pancreatic, adrenocortical, pituitary, and muscle functions. The attending physician utilized the Common Criteria for Toxicity (CTC)-AE version 5.0 to define and grade irAEs at each treatment cycle. The evaluation of irAEs was conducted without blinding due to the necessity of reporting to the pharmacovigilance agency. Following the Reviewer's recommendations, we have added a description of this standardized monitoring protocol to subsection “2.2 Data Collection and Evaluation of Results.”
Results and Statistical Analysis
Subgroup Analysis Power and Multiple TestingThe significant OS benefit in the high PD-L1 subgroup (TPS ≥50%, ICIs monotherapy) is intriguing but requires context. Was this subgroup pre-specified in the study hypothesis? Given the exploratory nature of this analysis, adjusting for multiple comparisons (e.g., Bonferroni correction) or reporting a false discovery rate would mitigate Type I error concerns. Additionally, clarify the sample size after PSM in this subgroup (e.g., n=? per arm).
- Subgroup analysis was a pre-specified secondary purpose of the study. In response to another Reviewer's comments, we described in detail the experimental hypothesis and parameters for calculating the study sample size. Nevertheless, we had no evidence to accurately estimate the sample size of subgroups based on the expression level of PD-L1 TPS. We clarified the sample size after PSM of these subgroups in subsection “3.3 Survival Analysis.”
- We mentioned alpha risk inflation from multiple comparisons as a major limitation of our study. In clinical study, one accept a pre-selected Type I error rate (alpha), i.e. false positive, of 5%. This error may be inflated (alpha inflation) by analyzing the same datum more than once. Alpha inflation refers to the phenomenon that the more statistical tests, the more likely one could find a significant result when it is actually not. There are essentially two common ways to deal with this type of problem. The first involves adjusting the type I error rate less than 0.05 by Bonferroni correction (Bland JM, Altman DG. Multiple significance tests: the Bonferroni method. BMJ 1995;310:170). Although Bonferroni adjustment is a simple technique to address the issue of alpha inflation, it is not without problem. Bonferroni adjustment is known for being too conservative and may increase the chance of higher than intended Type II error, i.e. false negative (Perneger TV. What’s wrong with Bonferroni adjustments. BMJ 1998;316:1236-8). The second involves the inclusion in the multivariate analysis of all covariates tested in the univariate analysis and not just those that showed a significant association with the intended outcome through the univariate testing (Rothman KJ. No adjustments are needed for multiple comparisons. Epidemiology 1990;1:43-6). We decided to adopt the latter methodology because the context in which we conducted our analysis was largerly unprecedented and excluding falsely negative prognostic factors could have undermined the reliability of the final results. We specified this methodological choice in the subsection “Statistical Analysis” and addressed this issue in more detail in the section on study limitations.
Exclusion of Short-Term ICI UsersPatients receiving <2 cycles of ICIs were excluded, which may introduce selection bias (these patients often have poor prognosis). Discuss how this exclusion might affect generalizability, particularly regarding safety outcomes (e.g., irAEs in frail populations).
- According to the Reviewer's suggestions, we emphasized more the implications of excluding patients receiving less than 2 cycles of treatment. These patients generally represent a frail population with a poor prognosis. Although the subgroup is small, their exclusion introduces a selection bias with a potential impact on the assessment of survival outcomes. The emerging evidence that a high frailty score might predict an increased risk of irAE implies an additional limitation that affects the generalizability of our findings [Olsson Ladjevardi C, et al. Predicting immune‐related adverse events using a simplified frailty score in cancer patients treated with checkpoint inhibitors: A retrospective cohort study. Cancer Med. 2023 Jun;12(12):13217-13224].
Heterogeneity of Booster TypesThe exposed cohort includes patients receiving 3rd dose, 4th dose, or bivalent boosters. Given potential differences in immunogenicity between booster types, stratified analyses (e.g., 3rd vs. 4th dose) or a sensitivity analysis addressing this heterogeneity would be valuable. If underpowered, this limitation should be explicitly acknowledged.
- The latter is a very intriguing observation that we honestly had not considered. However, the numerical distribution of patients makes this subgroup statistic underpowered and therefore unreliable. We made explicit reference to this issue in the section on the limitations of the study.
Discussion and Interpretation
Mechanistic Hypotheses for PD-L1 Subgroup EffectThe proposed mechanism linking mRNA vaccination to PD-L1 upregulation and enhanced ICI response is plausible but requires stronger mechanistic support. Cite preclinical studies showing mRNA vaccine-induced PD-L1 expression in antigen-presenting cells or tumor cells (e.g., studies on TLR activation or IFN-γ signaling).
- The mechanistic hypothesis of a direct link between COVID-19 mRNA vaccination and increased PD-L1 expression within the tumor microenvironment is clearly intriguing. We have already referred to all available evidence related to COVID-19 mRNA vaccination that could provide a rationale to support this hypothesis. However, following the Reviewer's suggestions, we have introduced new experimental data in the “Discussion” section that were not available at the time of the initial drafting. These data have shown that intratumoral injections of tozinameran induce abundant tumor-infiltrating lymphocytes with enhanced local levels of proinflammatory markers and significantly increased expression of PD-L1 on tumor-associated immune cells. These effects would result in changing the tumor immune microenvironment to a state more favorable for the therapeutic efficacy of immune checkpoint blockade [Li R, et al. The guided fire from within: intratumoral administration of mRNA-based vaccines to mobilize memory immunity and direct immune responses against pathogen to target solid tumors. Cell Discov. 2025 Jan 2;10:127].
Comparison with Prior StudiesThe manuscript notes discrepancies with retrospective studies (e.g., Grippin et al.), which may relate to differences in vaccination timing (booster vs. primary series) or baseline immunity (all patients here had prior 2-dose priming). Elaborate on these contextual differences to provide a more nuanced discussion.
- We do agree with the Reviewer that a more comprehensive comparative evaluation of our data with retrospective studies by Grippin and colleagues would be advisable. However, both referenced studies are currently only accessible in abstract form. This limitation hinders our ability to thoroughly examine the methodological differences and similarities in survival outcomes. We feel that any further interpretation, beyond the discussion points we have already addressed, would be tentative and overly speculative.
Cautiousness in ConclusionWhile the PD-L1 subgroup finding is hypothesis-generating, it should be explicitly framed as exploratory rather than definitive. Emphasize the need for prospective validation in randomized controlled trials (RCTs), ideally with correlative biomarker analyses.
- Accordingly, we emphasized the real experimental value of our findings. Their value as hypothesis-generating results mandates prospective validation in independent series and, ideally, in randomized controlled trials with correlative biomarker analyses.
Other Recommendations
Data TransparencyInclude a table of standardized differences post-PSM to demonstrate balance of covariates (e.g., ECOG PS, metastatic sites) between cohorts, as recommended in observational research reporting guidelines (e.g., STROBE).
- To further demonstrate the balance of covariates after PSM, we included a column in Table 1 showing the standardized mean difference (SMD) values. Then, we made consistent updates to the “Statistical Analysis” section and the reference list.
Update Reference CitationsReplace preprint citations (e.g., Ravera et al., 2024) with peer-reviewed publications where possible, or justify the use of preprints in the context of rapidly evolving evidence.
- The paper by Ravera et al. (2024) is still available only as preprints. We have justified the citation of this article using the format suggested by the Reviewer.
Guideline AlignmentDiscuss how these findings inform current vaccination guidelines for cancer patients (e.g., ASCO 2024, NCCN 2024), highlighting the safety profile and potential subgroup benefits to support personalized vaccination strategies.
- Our findings are consistent with the recommendations of relevant guidelines for vaccination of cancer patients on active treatment and do not suggest a personalized immunization strategy based on different safety and efficacy profiles (COVID-19 Vaccines in People with Cancer. https://www.cancer.org/cancer/managing-cancer/coronavirus-covid-19-and-cancer/covid-19-vaccines-in-people-with-cancer. [Accessed May 28, 2025]). We confirm that the primary rationale for recommending mRNA vaccination in advanced NSCLC patients eligible for immune checkpoint blockade is prevention of severe COVID-19.
Reviewer 2 Report
Comments and Suggestions for Authors
This manuscript presents a well-designed, prospective, real-world study evaluating the impact of periodic COVID-19 mRNA vaccine booster on the safety and efficacy of immune checkpoint inhibitors in patients with NSCLC. The findings are clinically relevant. However, minor revisios are recommended to enhace the clarity:
- Error en table 1: In Table 1 (PSM-adjusted population), the percentages in the "Exposed cohort" column for the "ECOG PS" row appear incorrect. Specifically, the values listed (e.g., 17.6% for ECOG PS 0, 67.6% for ECOG PS 1, 14.7% for ECOG PS 2) do not align with the expected distribution based on the sample size (N=102). Please recalculate and correct these percentages to ensure accuracy.
- Discussion of long-term toxicities: the discussion section thoroughly address the immediate safety profile of vaccine boosters with respect to irAEs. However, it would be benefit from a brief discussio of potential long-term toxicities, particularly given the immunomodulatory effects of both mRNA vaccines and ICIs.
- Figure legends: the legends for figures 1 and 2 are concise but could be expanded to include the statistical methos used (e.g. log-rank test for survival comparisons).
- Typographical note: on page 6, lines 305, the phrase "the current was an observational study" should be revised to "the current study was observational".
Author Response
Response to Reviewer 2 Comments
Comments and Suggestions for Authors
This manuscript presents a well-designed, prospective, real-world study evaluating the impact of periodic COVID-19 mRNA vaccine booster on the safety and efficacy of immune checkpoint inhibitors in patients with NSCLC. The findings are clinically relevant. However, minor revisios are recommended to enhace the clarity:
Error en table 1: In Table 1 (PSM-adjusted population), the percentages in the "Exposed cohort" column for the "ECOG PS" row appear incorrect. Specifically, the values listed (e.g., 17.6% for ECOG PS 0, 67.6% for ECOG PS 1, 14.7% for ECOG PS 2) do not align with the expected distribution based on the sample size (N=102). Please recalculate and correct these percentages to ensure accuracy.
- We completely revised the absolute values and percentages reported in Table 1 and corrected the calculation errors pointed out by the Reviewer.
Discussion of long-term toxicities: the discussion section thoroughly address the immediate safety profile of vaccine boosters with respect to irAEs. However, it would be benefit from a brief discussio of potential long-term toxicities, particularly given the immunomodulatory effects of both mRNA vaccines and ICIs.
- Our assessment was thorough in addressing immune-related toxicities during the course of treatment, with an observation period of approximately six months. Although they are less frequent, our safety monitoring may not have detected long-term irAEs as accurately. We made explicit reference to this possibility in the “Discussion” section.
Figure legends: the legends for figures 1 and 2 are concise but could be expanded to include the statistical methos used (e.g. log-rank test for survival comparisons).
- We have added the statistical methods used for comparing survival and calculating HR with the corresponding 95% CI in all panels of Figures 1 and 2.
Typographical note: on page 6, lines 305, the phrase "the current was an observational study" should be revised to "the current study was observational".
- We corrected this statement based on the Reviewer's suggestion.
Reviewer 3 Report
Comments and Suggestions for Authors
MAJOR POINTS
1. EXPECTED INFLUENCE OF VACCINES ON ICI EFFICACY: SUPERIORITY VS NON-INFERIORITY VS NO PRE-SPECIFIED HYPOTHESIS
a) The evidence on this topic published so far is, in some cases, presented with insufficient detail, and, in other cases, through a contradictory approach. This issue arises early in the abstract, where the authors state that vaccines might influence the efficacy of ICIs”. The controversy lies in the 'direction' of this influence, as the wording of the entire paragraph is open to two interpretations: vaccines could enhance the efficacy of ICIs, or they could reduce it. This doubt should be avoided in readers.
b) In more detail, if vaccines are expected to enhance the efficacy of ICIs, the analysis should be based on a superiority design. Conversely, if vaccines could potentially reduce the efficacy of ICIs, a non-inferiority design probably would be more appropriate. While one cannot exclude the possibility that both hypotheses deserve to be evaluated (which would represent a third hypothesis), if this is the case, it should be stated more clearly.
2. THE PRESENTATION OF THE PREVIOUS LITERATURE IS EXCESSIVELY DIVIDED BETWEEN THE INTRODUCTION AND THE DISCUSSION, THUS RAISING THE RISK OF PROVIDING AN UNCLEAR SUMMARY OF THE CURRENT EVIDENCE AT THE BEGINNING OF THE ARTICLE.
a) In the Introduction, the paragraph emphasising that “... expert committees advocate for the regular administration of updated SARS-CoV-2 vaccines to cancer patients receiving immunosuppressive therapies” is correctly placed at the beginning. This is correctly placed at the beginning.
b) Lines 91–105 (including references 9–17): the presence of “a complex interplay” is undeniable, as is the fact that “these interactions raise concerns about potential clinical consequences, including the intensification of immune-related adverse events (irAEs), and the impact on vaccine efficacy and cancer treatment outcomes [14]”. However, the purpose of this investigation remains unclear up to the end of the Introduction, as confirmed by Point 1 regarding the abstract.
c) On the other hand, the beginning of the Discussion (see lines 285–286, along with reference 26) finally clarifies that the authors' main hypothesis is that “the simultaneous administration of mRNA vaccines and ICIs has attracted considerable attention due to the potential to enhance immune responses (26).”
d) Lines 285–295: References 27–33 are extremely important in the context of the article. However, in my view, these references (along with the potential conclusions they suggest) should have been cited in the Introduction. As a rough suggestion, lines 285–301 could be moved from the Discussion to the Introduction.
3. RESULTS SECTION
a) While the decision to use propensity-matching statistics (with a 1:1 ratio) is appropriate, the results generated by this method are somewhat surprising. The usefulness of these statistics is maximised when the PM process excludes a large number of patients, resulting in two final populations with an equal number of patients that are much more closely matched than the initial two populations. In this analysis, however, the improvement in the degree of matching is only modest because the initial two populations (reference cohort: N = 114; exposed cohort: N = 112, totalling 226 patients) are not particularly smaller than the two PM populations (reference cohort: N = 102; exposed cohort: N = 102, totalling 204 patients).
b) Presenting the results in a way that gives the same degree of priority to both the original population and the PM population (e.g. Table 1) makes the presentation unnecessarily complex. Which results did this study generate? Those of the original population or the PM population?
c) I suggest that the authors make a clear choice on this point. One suggestion could be to move one of these two analyses (at the authors’ discretion) from the Results section to a new Appendix. I believe this decision could greatly enhance the clarity of the entire article.
4. SPECIFIC POINTS:
a) In the legends to Tables 4 and 5, please report the size of the overall population explicitly, as in Tables 1 and 2.
b) In both panels of Figure 1 (or in the legend to Figure 1), please report the hazard ratio value (together with its 95% confidence interval).
c) The authors should emphasise more explicitly somewhere in the paper that the exposed cohort showed a numerical survival advantage in both Kaplan-Meier plots in Figure 1, even though this difference remained far from reaching statistical significance (or very far for overall survival). If the authors agree, this suggests that the curves of the exposed cohort are non-inferior to some extent, although one should bear in mind that no non-inferiority analysis was performed. Reporting the HR values is helpful in establishing whether this observation is valid.
Author Response
Response to Reviewer 3 Comments
Comments and Suggestions for Authors
MAJOR POINTS
- EXPECTED INFLUENCE OF VACCINES ON ICI EFFICACY: SUPERIORITY VS NON-INFERIORITY VS NO PRE-SPECIFIED HYPOTHESIS
- a) The evidence on this topic published so far is, in some cases, presented with insufficient detail, and, in other cases, through a contradictory approach. This issue arises early in the abstract, where the authors state that vaccines might influence the efficacy of ICIs”. The controversy lies in the 'direction' of this influence, as the wording of the entire paragraph is open to two interpretations: vaccines could enhance the efficacy of ICIs, or they could reduce it. This doubt should be avoided in readers.
- We sincerely appreciate these insightful remarks, which reflect a thorough evaluation of our work and a comprehensive understanding of the subject matter. We acknowledge that the introduction's wording might not have clearly conveyed the focus and intent of our analysis. In response to the subsequent comments, we have revised the latter portion of the "introduction" section to ensure that the analysis's objective is clearly and easily understood.
- b) In more detail, if vaccines are expected to enhance the efficacy of ICIs, the analysis should be based on a superiority design. Conversely, if vaccines could potentially reduce the efficacy of ICIs, a non-inferiority design probably would be more appropriate. While one cannot exclude the possibility that both hypotheses deserve to be evaluated (which would represent a third hypothesis), if this is the case, it should be stated more clearly.
- Our analysis was based on a superiority design for two main reasons. Firstly, the available evidence suggests that COVID-19 mRNA vaccination may improve the efficacy of immune checkpoint blockade. The latter consideration, together with the lack of experimental data demonstrating the contrary hypothesis, makes a presumed superiority design more plausible than non-inferiority. Secondly, although the third hypothesis cannot reasonably be ruled out, it would require a much larger sample to be proven and goes beyond the scope of exploratory research as the present one has the presumption to be. In agreement with the present suggestion and in response to another Reviewer's comments, we explicitly described the purposes of the study's experimental hypothesis in the “Statistical analysis” section.
- THE PRESENTATION OF THE PREVIOUS LITERATURE IS EXCESSIVELY DIVIDED BETWEEN THE INTRODUCTION AND THE DISCUSSION, THUS RAISING THE RISK OF PROVIDING AN UNCLEAR SUMMARY OF THE CURRENT EVIDENCE AT THE BEGINNING OF THE ARTICLE.
- a) In the Introduction, the paragraph emphasising that “... expert committees advocate for the regular administration of updated SARS-CoV-2 vaccines to cancer patients receiving immunosuppressive therapies” is correctly placed at the beginning. This is correctly placed at the beginning.
- b) Lines 91–105 (including references 9–17): the presence of “a complex interplay” is undeniable, as is the fact that “these interactions raise concerns about potential clinical consequences, including the intensification of immune-related adverse events (irAEs), and the impact on vaccine efficacy and cancer treatment outcomes [14]”. However, the purpose of this investigation remains unclear up to the end of the Introduction, as confirmed by Point 1 regarding the abstract.
- c) On the other hand, the beginning of the Discussion (see lines 285–286, along with reference 26) finally clarifies that the authors' main hypothesis is that “the simultaneous administration of mRNA vaccines and ICIs has attracted considerable attention due to the potential to enhance immune responses (26).”
- d) Lines 285–295: References 27–33 are extremely important in the context of the article. However, in my view, these references (along with the potential conclusions they suggest) should have been cited in the Introduction. As a rough suggestion, lines 285–301 could be moved from the Discussion to the Introduction.
- We do agree with the Reviewer that the references supporting the experimental hypothesis were inadequately distributed. The citation of available evidence indicating a potential enhancement effect of COVID-19 mRNA vaccination on immune checkpoint blockade should not have been included in the early part of the introduction. The previous arrangement of both the introduction and the discussion might indeed be unclear and perplexing to readers. Consequently, we have thoroughly revised these sections in line with the Reviewer's recommendations. Nonetheless, we opted to retain the reference to the pivotal data from the Vax-On-Third study (previous reference 33) at the beginning of the discussion, as we consider it a fundamental point for developing subsequent arguments.
- RESULTS SECTION
- a) While the decision to use propensity-matching statistics (with a 1:1 ratio) is appropriate, the results generated by this method are somewhat surprising. The usefulness of these statistics is maximised when the PM process excludes a large number of patients, resulting in two final populations with an equal number of patients that are much more closely matched than the initial two populations. In this analysis, however, the improvement in the degree of matching is only modest because the initial two populations (reference cohort: N = 114; exposed cohort: N = 112, totalling 226 patients) are not particularly smaller than the two PM populations (reference cohort: N = 102; exposed cohort: N = 102, totalling 204 patients).
- We confirm that employing PSM statistics is essential in observational clinical trials to lessen the impact of selection bias. When applied to case series with a notable imbalance of prognostic covariates at baseline, PSM generally leads to a considerable reduction in sample size. This exclusion of a large number of cases is primarily observed in observational studies where patient selection and cohort definition are determined retrospectively. In our case series, using PSM led to a slight reduction in sample size (approximately 10%) for several reasons. The enrollment of patients was done prospectively, adhering strictly to inclusion and exclusion criteria set by the government agency's prescribing and reimbursement guidelines. This method of recruitment resulted in highly homogeneous case series with minimal differences, nearly reaching statistical significance, in the distribution of only two covariates. In this scenario, it is conceivable and not surprising that the PSM algorithm resulted in only a slight decrease in the numerical composition of the cohorts.
- b) Presenting the results in a way that gives the same degree of priority to both the original population and the PM population (e.g. Table 1) makes the presentation unnecessarily complex. Which results did this study generate? Those of the original population or the PM population?
- We described the original raw population only to motivate the application of PSM. To reduce interpretive complexity, we clearly stated that only the population adjusted for PSM was relevant to all subsequent evaluations.
- c) I suggest that the authors make a clear choice on this point. One suggestion could be to move one of these two analyses (at the authors’ discretion) from the Results section to a new Appendix. I believe this decision could greatly enhance the clarity of the entire article.
- Following the Reviewer's suggestions, we moved the description of unadjusted population characteristics to the Supplementary Materials (now cited as Supplementary Table S1) and retained only the description of PSM-adjusted population characteristics (now cited as Table 1) in the main text.
- SPECIFIC POINTS:
- a) In the legends to Tables 4 and 5, please report the size of the overall population explicitly, as in Tables 1 and 2.
- We added the relevant population for the analysis (PSM-adjusted population) and relative size in the heading of each table.
- b) In both panels of Figure 1 (or in the legend to Figure 1), please report the hazard ratio value (together with its 95% confidence interval).
- We reported the value of the hazard ratio (along with its 95% confidence interval) in both panels of Figure 1.
- c) The authors should emphasise more explicitly somewhere in the paper that the exposed cohort showed a numerical survival advantage in both Kaplan-Meier plots in Figure 1, even though this difference remained far from reaching statistical significance (or very far for overall survival). If the authors agree, this suggests that the curves of the exposed cohort are non-inferior to some extent, although one should bear in mind that no non-inferiority analysis was performed. Reporting the HR values is helpful in establishing whether this observation is valid.
- We modified the relevant part of the “Discussion” section according to the Reviewer's suggestions as follows. Regarding the primary purpose of this research, survival analysis in the general population revealed no differences between the cohorts in terms of PFS or OS. Even though the exposed cohort showed a numerically longer duration for both PFS and OS, the prolongation of survival remained far from reaching statistical significance. These findings do not support the experimental hypothesis of potential superiority associated with periodic exposure to COVID-19 vaccination and instead indicate a non-inferiority outcome.
Round 2
Reviewer 1 Report
Comments and Suggestions for Authors
The revised manuscript successfully addresses the majority of prior critiques, offering a rigorous analysis of COVID-19 vaccine boosters in advanced NSCLC. The clarification of methodological details, enhanced mechanistic discussions, and transparent acknowledgment of limitations strengthen the study’s validity. While minor issues persist, they do not significantly undermine the core findings. The work provides valuable evidence supporting the safety of boosters in this population, with intriguing signals for benefit in high PD-L1 subsets.
Remaining Limitations and Suggestions
- Immortal-Time Bias Residuals: Despite the predefined vaccination window, there may still be residual bias from patients who delayed ICI initiation after vaccination. The authors could further address this by analyzing time-to-treatment initiation between cohorts.
- Long-Term irAE Data: The median follow-up for irAEs is 5.6 months, which may miss late-onset toxicities. A note on the need for longer-term safety surveillance would be prudent.
- Guideline Alignment: While the conclusion mentions ASCO and NCCN guidelines, a direct comparison of study findings with recent recommendations (e.g., ASCO 2024 on cancer vaccination) would enhance clinical relevance.
Author Response
Response to Reviewer 1 Comments (second round)
Comments and Suggestions for Authors
The revised manuscript successfully addresses the majority of prior critiques, offering a rigorous analysis of COVID-19 vaccine boosters in advanced NSCLC. The clarification of methodological details, enhanced mechanistic discussions, and transparent acknowledgment of limitations strengthen the study’s validity. While minor issues persist, they do not significantly undermine the core findings. The work provides valuable evidence supporting the safety of boosters in this population, with intriguing signals for benefit in high PD-L1 subsets.
Remaining Limitations and Suggestions
Immortal-Time Bias Residuals: Despite the predefined vaccination window, there may still be residual bias from patients who delayed ICI initiation after vaccination. The authors could further address this by analyzing time-to-treatment initiation between cohorts.
- We appreciated this insightful comment once again. Although the vaccination window relied on the results of previous research and immunological criteria, the duration of 90 is really quite lengthy. Compared to existing evidence, the predefined setting of our framework certainly reduces the immortal-time bias. However, as suggested by the Reviewer, we must acknowledge that the time frame between vaccination and treatment represents a residual confounding. Within the exposed cohort, 13 patients (12.7%) received the booster vaccination after starting ICI therapy, with a median delay of 2 days (ranging from 1 to 6 days). The remaining 89 patients (87.3%) were given the booster vaccination before initiation of treatment, with a median advance of 18 days (ranging from 1 to 50 days). We have added this additional data at the end of subsection “3.1 Patient characteristics” and specifically mentioned the latter issue among the limitations of the study.
Long-Term irAE Data: The median follow-up for irAEs is 5.6 months, which may miss late-onset toxicities. A note on the need for longer-term safety surveillance would be prudent.
- Our assessment was thorough in addressing immune-related toxicities during the course of treatment, with an observation period of approximately six months. Although they are less frequent, our safety monitoring may not have detected long-term irAEs as accurately. The latter consideration warrants caution and the need for further surveillance concerning late-onset adverse events. These findings suggest that recommended booster doses of COVID-19 mRNA vaccines, given shortly before or after beginning treatment, do not influence the efficacy and short-term safety of immune checkpoint blockade. We have modified the specific part of the discussion accordingly.
Guideline Alignment: While the conclusion mentions ASCO and NCCN guidelines, a direct comparison of study findings with recent recommendations (e.g., ASCO 2024 on cancer vaccination) would enhance clinical relevance.
- In agreement with the latest advances and recommendations, the primary rationale for advocating mRNA vaccination in advanced NSCLC patients eligible for immune checkpoint blockade is the prevention of severe COVID-19 [Kamboj, M.; et al. More Frequent than Annual Administration of COVID-19 Vaccination May Be Appropriate for Patients With Cancer. JCO Oncology Adv. 2025, 2:e2400107].
Reviewer 3 Report
Comments and Suggestions for Authors
The authors have satisfactorily addressed all of the points I raised in my review.
Author Response
Response to Reviewer 3 Comments (second round)
Comments and Suggestions for Authors
The authors have satisfactorily addressed all of the points I raised in my review.
- We sincerely appreciated the insightful comments from the previous round. The queries and recommendations raised demonstrate a thorough understanding of the subject matter and clinical research methodology. The Reviewer's suggested changes have improved the methodological approach of the study and the interpretation of the results.